# Tracking-forced Referring Video Object Segmentation

Ruxue Yan
VCIP, TMCC, TBI Center
College of Computer Science
Nankai University
Tianjin, China
yanruxue@dbis.nankai.edu.cn

Wenya Guo*
VCIP, TMCC, TBI Center
College of Computer Science
Nankai University
Tianjin, China
guowenya@dbis.nankai.edu.cn

Xubo Liu
VCIP, TMCC, TBI Center
College of Computer Science
Nankai University
Tianjin, China
liuxubo@dbis.nankai.edu.cn

Xumeng Liu
VCIP, TMCC, TBI Center
College of Computer Science
Nankai University
Tianjin, China
liuxumeng@dbis.nankai.edu.cn

Ying Zhang
VCIP, TMCC, TBI Center
College of Computer Science
Nankai University
Tianjin, China
yingzhang@nankai.edu.cn

Xiaojie Yuan
VCIP, TMCC, TBI Center
College of Computer Science
Nankai University
Tianjin, China
yuanxj@nankai.edu.cn

## ABSTRACT

Referring video object segmentation (RVOS) is a cross-modal task that aims to segment the target object described by language expressions. A video typically consists of multiple frames and existing works conduct segmentation at either the clip-level or the frame-level. Clip-level methods process a clip at once and segment in parallel, lacking explicit inter-frame interactions. In contrast, frame-level methods facilitate direct interactions between frames by processing videos frame by frame, but they are prone to error accumulation. In this paper, we propose a novel tracking-forced framework, introducing high-quality tracking information and forcing the model to achieve accurate segmentation. Concretely, we utilize the ground-truth segmentation of previous frames as accurate inter-frame interactions, providing high-quality tracking references for segmentation in the next frame. This decouples the current input from the previous output, which enables our model to concentrate on accurately segmenting just based on given tracking information, improving training efficiency and preventing error accumulation. For the inference stage without ground-truth masks, we carefully select the beginning frame to construct tracking information, aiming to ensure accurate tracking-based frame-by-frame object segmentation. With these designs, our tracking-forced method significantly outperforms existing methods on 4 widely used benchmarks by at least 3%. Especially, our method achieves 88.3% P@0.5 accuracy and 87.6 overall IoU score on the JHMDB-Sentences dataset, surpassing previous best methods by 5.0% and 8.0, respectively.

## CCS CONCEPTS

• **Computing methodologies → Video segmentation**.

*Corresponding author.

## KEYWORDS

Referring video object segmentation, Tracking-forced framework, Parallel training, Sequential inference

**ACM Reference Format:**
Ruxue Yan, Wenya Guo, Xubo Liu, Xumeng Liu, Ying Zhang, and Xiaojie Yuan. 2024. Tracking-forced Referring Video Object Segmentation. In *Proceedings of the 32nd ACM International Conference on Multimedia (MM '24), October 28-November 1, 2024, Melbourne, VIC, Australia.* ACM, New York, NY, USA, 10 pages. https://doi.org/10.1145/3664647.3680817

## 1 INTRODUCTION

The purpose of Referring Video Object Segmentation (RVOS) is to accurately segment the target object that is referred to by a natural language expression in each frame. This emerging task has garnered significant attention owing to its catalytic effect on various fields, such as video editing [2, 12], self-driving vehicle [10, 13]. Since RVOS requires to identify the same target object from all frames of a video, it is necessary not only to comprehensively understand the cross-modal sources, *e.g.*, vision and language, but also to effectively track the same object across frames, which is particularly crucial.

Existing works for the RVOS task can be divided into two categories based on their different video processing units: clip-level methods [1, 8, 29] and frame-level methods [11, 25, 28]. As shown in Fig.1(a), clip-level methods process the frames of a clip in a parallel way. These works typically utilize feature association [8] or query sharing [1, 29] techniques to track the same object across frames, lacking explicit inter-frame interactions modeling. Frame-level methods utilize inter-frame interaction information to assist with more accurate object tracking. As illustrated in Fig.1(b), these methods process the video frame by frame, in which each frame utilizes the segmentation result of its previous frame as guidance to assist in tracking target objects in the current frame. This approach seems reasonable for tracking the target object, but it may lead to error accumulation. This is because the information transmitted from the previous frame is not entirely accurate, and the inaccurate information can affect tracking and segmenting the referred objects in the next frame. Furthermore, the frame-by-frame processing method also decreases the training efficiency.

To address above limitations of existing methods, we propose a novel **T**racking-**F**orced **F**ramework (**TF**²) for the RVOS task. TF² is

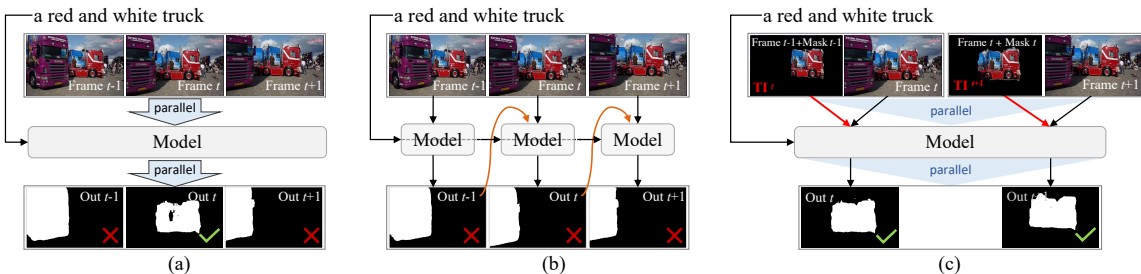

Figure 1: Illustration of different frame processing methods: (a) clip-level methods, (b) frame-level methods, and (c) our tracking-forced method which introduces high-quality tracking information (TI) derived from the previous frame and the previous ground-truth mask. The segmentation results in (a) and (b) are from Referformer[29] and OnlineRefer[28], respectively.

motivated by the observation that the referred object's motion trajectory is uninterrupted along with the interaction among frames. Aforementioned frame-level methods track this information by linking previous outputs to current inputs, often suffering from error accumulation due to the use of inaccurate tracking information. To provide as precise tracking information as possible, $TF^2$ is designed to introduce high-quality tracking information and force the model to learn from it, thereby achieving more accurate segmentation. Specifically, as shown in Fig.1(c), we explicitly utilize ground-truth masks to compute accurate tracking information, obtaining valid appearance and location of the object in the previous frame, which serves as the reference for the current frame segmentation process. Under the guidance of the constructed precise tracking information, our model is forced to concentrate on accurately segmenting, which is the ultimate goal of this task. Besides, by decoupling the previous output and the current input, we can achieve a parallel training process, which is more efficient than previous frame-level works.

After the above training process, our model can learn to segment the target object in a frame with the guidance of accurate tracking information. When it comes to the inference process where ground-truth labels are unavailable, we can not directly obtain tracking information for all frames at once. To reduce this gap between the training and inference process, we gradually construct tracking information based on predicted segmentation results. Note that we start the inference process from a carefully selected key frame, rather than simply beginning with the first frame as existing frame-level methods do. The chosen key frame should meet the criterion of displaying the target object as completely as possible. Benefiting from the reliability of the well-chosen key frame, accurate tracking information is provided to help with precise segmentation, and the issue of error accumulation present in existing frame-level methods is significantly mitigated.

In summary, our contributions are:

- We propose a tracking-forced framework for RVOS, utilizing high-quality tracking information to focus the model on segmentation. Our method is superior to clip-level methods by integrating inter-frame interactions and boasts higher training efficiency than frame-level methods.
- Ground-truth masks are fully utilized to provide completely accurate tracking inference for object segmentation, improving the training effectiveness by parallel processing. The inference process is started from a carefully selected key

frame to mitigate the error accumulation that widely existed in previous frame-level works.
- Extensive experiments are conducted on Ref-Youtube-VOS, Refer-DAVIS17, A2D-Sentences, and JHMDB-Sentences. Our method outperforms all previous methods and achieves state-of-the-art performance.

## 2 RELATED WORKS

### 2.1 Referring Video Object Segmentation

Referring Video Object Segmentation (RVOS) aims to accurately segment objects in video clips based on textual descriptions. Early works [17, 21, 23, 25–27] relied on complex network structures to align multimodal information for object tracking and segmentation. For example, URVOS [25] presents an end-to-end deep neural network that accomplishes both language-based object segmentation and mask propagation in a unified model. [8] proposes a collaborative spatial-temporal framework that integrates temporal information for action recognition and spatial information for accurate actor segmentation. Moreover, MTTR [1] and ReferFormer [29] propose transformer-based end-to-end framework during the same period. To foster the RVOS, [28] propose a solid online framework based on query propagation. TCE-RVOS [7] achieves state-of-the-art performance by effectively learning temporal information.

Compared to existing SOTA methods, our approach introduces a tracking-forced framework. This innovative inclusion enables parallel training, eliminating the dependence on frame-level propagation. Consequently, our method exhibits a substantial enhancement in training efficiency.

### 2.2 Object Tracking Methods in RVOS

The core of the RVOS lies in the localization and tracking of target objects. For the former, current methods have achieved impressive performance in accurately localizing objects, leveraging powerful visual backbones such as ResNet [5] and Video-Swin [20]. However, for the latter, objects tend to exhibit motion between frames, and multiple instances of the same object may be present, posing challenges for accurate tracking of the queried object and thus limiting segmentation performance. Existing approaches [28] achieve simple tracking by query propagation from one frame to the next. This simple tracking method is prone to error accumulation (*e.g.*, the referred object does not appear in the first frame or appears incompletely) thus leading to tracking in the wrong direction. To

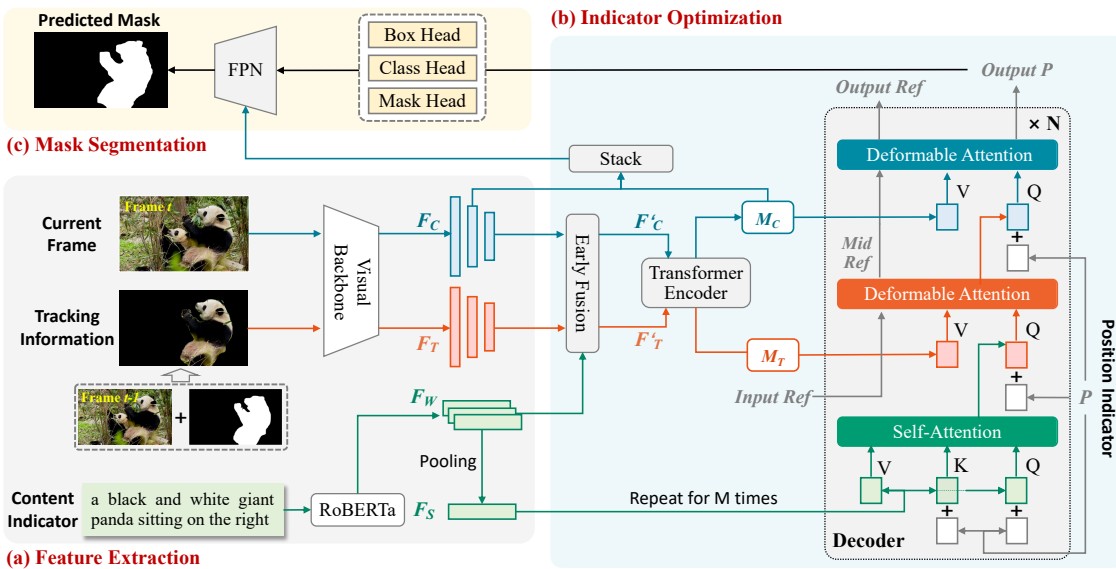

**Figure 2: The overall architecture of our proposed tracking-forced framework. Given the current frame to be segmented, we first extract features of the current frame and its tracking information, then conduct indicator optimization to retrieve the referred object, and finally obtain the predicted mask. "Ref" denotes the reference points and "P" represents position indicators.**

tackle this, we propose a tracking-forced segmentation strategy to ensure the tracking direction for better segmentation.

## 3 METHOD

We propose a tracking-forced framework, introducing tracking information during the segmentation process and forcing the model to learn visual information about target objects from it. In this section, we present our model architecture and our training-inference approach. Compared to previous works for RVOS where model training and inference are conducted in the same way, we perform different forms for training and inference. We first train our TF$^2$ in a parallel way and then conduct sequential inference. During the parallel training process in Sec.3.1, we utilize ground-truth masks to construct tracking information, which is used with the referring language expression together for retrieving the target object. At the inference stage in Sec.3.2, we no longer start inference from the first frame of a video like previous frame-level methods but choose a reliable key frame as the beginning based on semantic similarities between all frames and the language expression. As illustrated in Fig.2, our model accepts the current frame, its corresponding tracking information, a language expression as input, and outputs the predicted mask without post-process. The implementation details are stated as follows.

### 3.1 Indicator-optimized Parallel Training

Usually, ground-truth masks are just used to calculate the training loss in previous works. In our TF$^2$, we further utilize ground-truth masks to construct semantically rich tracking information, which can provide visual information of referred objects for segmenting.

*3.1.1 Feature Extraction.* Given a frame to be segmented, we first construct its tracking information. Tracking information is essentially derived from the interactions between the current frame and

its adjacent frames. So we randomly select its previous or following frame as its adjacent frame used for tracking information construction. To obtain explicit visual information about the referred object, we perform an AND operation on RGB values of the adjacent frame based on its binary ground-truth mask. In this way, we can get the tracking information in which only the target object is retained. Then we adopt visual backbones (ResNet-50 and Swin-Transformer) to extract feature maps for the current frame and its tracking information, resulting in visual features $F_C$ and $F_T$, respectively. As for the language expression with the $L$ words, we use RoBERTa [18] to extract the word-level text features $F_w = \{f_t^w\}_{t=1}^L$ and also pool the features of each word to obtain the sentence-level feature $F_s$.

*3.1.2 Indicator Optimization.* After obtaining the feature embedding of visual and text input, we follow the architecture of the Deformable DETR detector [35] for referred object detection. We mainly modify the decoder mechanism of the Deformable DETR by introducing a tracking-forced attention module, which is beneficial for accurate tracking.

We first prepare for the input of the decoder. We map the obtained current frame feature $F_C$ and tracking feature $F_T$ into the dimension $C = 256$. We then conduct an attention-based early-fusion between the word-level text feature $F_W$ and $F_C$, $F_T$ separately to enrich the visual information before the Transformer encoder layers. Among them, the text feature $F_w$ serves as Query, and the visual features serve as Key, Value, ultimately resulting in new feature maps $F_C'$, $F_T'$. The semantic features are fed into the Transformer encoder. Finally, the encoded memory of the current frame and tracking information, *i.e.*, $M_C$ and $M_T$ are input to the decoder.

In the decoding stage, to distinguish object queries from the query concept in the attention mechanism, we refer to object queries as object indicators in the following. Object indicators are

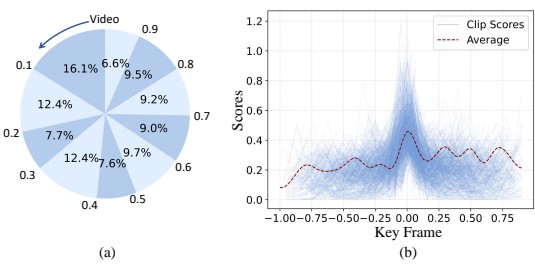

(a)                                (b)

**Figure 3: (a) The distribution of key frames in each stage of videos, and (b) statistics of CLIP scores of each frame relative to the key frames.**

composed of two parts: the content indicator and the position indicator. The content indicator is actually the sentence-level feature of the referred language expression, querying for object instances in videos. At the same time, the position indicator restores location information, which is also required to segment the referred object. We initialized the position indicator randomly and set its number as $M$. To fuse the above two, we also repeat the content indicator $F_S$ for $M$ times to fit the number of position queries. In each decoder layer, we perform a self-attention module and two deformable attention modules to optimize object indicators. A self-attention on object indicators is conducted at first to clearly define the information each object indicator concentrates on, in which only content indicators serve as Value and the whole object indicators serve as Query and Key. In the following deformable module, we adopt a tracking-forced approach to guide object indicators' optimization through tracking information. Reference points are injected for constructing Key in this attention processing. They are initially mapped from object indicators and used to extract only the features of specific areas related to object indicators as Key, which helps to improve the convergence speed of the model. Noticed that the tracking information contains explicit visual content of the referred object, object indicators are no longer conditioned solely on a language expression, but also incorporate the bounding box of the target object of its adjacent frame through this cross-attention. Input reference points are also updated as mid-reference points, which refer to areas more relevant to the target objects. Both the mid reference points and output object indicators through the learning from tracking information are fed into the next deformable attention module, in which $F_C$ serves as Value for target object detection. Object indicators and reference points are transmitted and optimized during the decoding process layer by layer. We adopt the output indicators of the last layer for mask segmentation.

*3.1.3 Mask Segmentation.* Based on the output object indicators of the last decoder layer, we use three lightweight heads on position indicators to get bounding boxes, categories, and corresponding masks of referred objects separately. Assuming the number of position indicators is set to $N$, let denote these head outputs as $\hat{y} = \{y_i\}_{i=1}^{N}$, which consists of three parts:

$$\hat{y}_i = \{\hat{c}_i, \hat{b}_i, \hat{m}_i\}, \tag{1}$$

where $\hat{c} \in \mathbb{R}^{N \times 1}$ represents the class categories of the retrieved objects, $\hat{b} \in \mathbb{R}^{N \times 4}$ is normalized vectors defining center coordinates

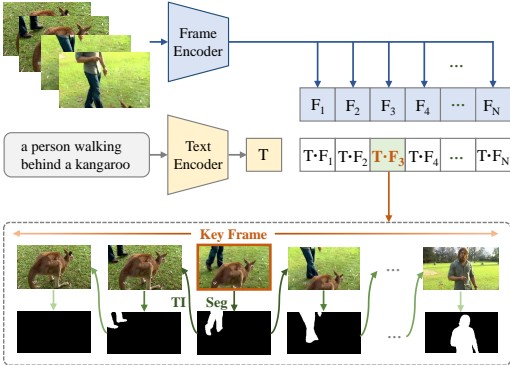

**Figure 4: Illustration of the inference process. The key frame is identified by matching video frames to a language expression with CLIP. This key frame serves as the starting point for our video segmentation inference. "TI" denotes the tracking information.**

as well as the width and height of predicted bounding boxes, $\hat{m} \in \mathbb{R}^{N \times \frac{H}{4} \times \frac{W}{4}}$ is the generated binary masks. We follow the matching approach to calculate the training loss. Considering that there is only one target object referred to by the language expression, we just need to minimize the matching loss between predictions and the ground-truth lables to find our positive sample:

$$\hat{y}_{\text{pos}} = \arg \min \mathcal{L}_{\text{match}}\{\hat{y}, y\}, \tag{2}$$

$$\mathcal{L}_{\text{match}} = \lambda_c \mathcal{L}_{\text{cls}} + \lambda_b \mathcal{L}_{\text{box}} + \lambda_m \mathcal{L}_{\text{mask}}, \tag{3}$$

where $\mathcal{L}_{\text{cls}}$ is the focal loss [16], $\mathcal{L}_{\text{box}}$ represents L1 loss and GIoU loss [32], and $\mathcal{L}_{\text{mask}}$ denotes the combination of DICE loss [22] and binary mask focal loss.

## 3.2 Key-frame-guided Sequential Inference

As there are no pre-existing ground-truth masks that can serve as direct tracking information during the inference process, we utilize a sequential inference approach to progressively construct tracking information and produce masks on a frame-by-frame basis. The architecture is the same as the training process shown in Fig.2. In this way, the key frame as the inference start is particularly important, as it affects the accuracy of subsequent tracking information.

*3.2.1 Key Frame Selection.* The frame chosen as the starting point for inference should meet the criterion of displaying the referred object as comprehensively as possible, ensuring ease of detection. Alternatively stated, it is ideal to rely solely on language expression-guided queries to accomplish the retrieval of the referred object in this fey frame. We utilize CLIP to find this reliable frame. For a video $\mathcal{I} = \{I_i\}_{i=1}^{T}$ with $T$ frames and a corresponding language expression $\mathcal{E}$, we calculate the semantic similarities between each frame and the referred expression. The frame with the highest similarity score is taken as the start point inference:

$$\sigma = \underset{i \in [1,T]}{\arg \max}(cosine\_similarity(I_i, \mathcal{E})), \tag{4}$$

where $\sigma \in [1, T]$ denotes the index of the key frame. In this way, we can obtain key frames of each referred language expression

and conduct statistical analysis on the position distribution of our selected key frames.

Take the validation set of Ref-Youtube-VOS as an example, we visualize the number and proportion of key frames located in each stage of videos in Fig.3(a). The key frame numbers of each stage in videos are equivalent. This verifies that due to the diversity of video content and language expressions, the positions of the most relevant frame to referred text in each video are not the same. So it is reasonable to choose a key frame as the inference start instead of always beginning at the first frame to mitigate the wide existing error accumulation. Furthermore, we align videos based on the key frames and calculate the average scores of each stage to intuitively represent the similarity scores distribution in Fig.3(b). The specific calculation methods for video alignment and relative scores calculation are detailed in the supplementary material. The key frames we selected have the highest semantic similarity with the referred text, which further proves that the reliability of our selected key frames as tracking starts. Noticed that other small peaks are appearing in the average curve, indicating that there are still frames highly relevant to the text, except for key frames. Based on this point, we also consider taking multiple key frames as starts to accelerate inference. We divide a video into equal clips at first and then choose the frame with the highest similarity score as the key frame of each clip. We adopt this selection approach instead of directly choosing the frames with top K scores because the highest frames are usually adjacent, which not can effectively help acceleration. The performance with different numbers of key frames is detailed in Sec.4.4.3.

*3.2.2 Spreading Inference.* After getting the reliable frame as tracking beginning, we could conduct a spreading reference to constructing tracking information frame by frame. During the spreading process, as shown in Fig.4, whenever we obtain the segmentation of a frame, we can construct the tracking information for the next frame, until the complete segmentation of the whole video. As for the key frame, we consider itself as its tracking information. In this way, for a video $I = \{I_i\}_{i=1}^{T}$ consisting $T$ frames with its key frame $I_\sigma$ is known, the Segmentation results $S$ can be obtained:

$$S = \{S_i\}_{i=1}^{T} = \begin{cases} TF^2(I_i, \mathcal{E}, S_{i+1}), & 1 \le i < \sigma, \\ TF^2(I_i, \mathcal{E}, I_i), & i = \sigma, \\ TF^2(I_i, \mathcal{E}, S_{i-1}), & \sigma < i \le T. \end{cases} \quad (5)$$

## 4 EXPERIMENTS

### 4.1 Experimental Settings

**Datasets.** We evaluate our method on four popular RVOS benchmarks. **Ref-Youtube-VOS** [25] is extended from the classic video object segmentation dataset Youtube-VOS [31] by introducing language expressions as reference. It contains a total of 3,978 videos (3,471 for training, 202 for validation, and 305 for testing) and 27,899 expressions. Each video is accompanied by one or more natural language expressions as references for segmenting objects in videos. **Ref-DAVIS17** [11] is an extension of another video segmentation dataset DAVIS17 [24], in the same form as Ref-Youtube-VOS, containing 90 videos and more than 1,500 expressions. **A2D-Sentences** [4] and **JHMDB-Sentences** [4] separately sourced from the original action and actor datasets A2D [30] and JHMDB [9]

with adding language expressions for segmenting. A2D-Sentences consists of 3,782 videos and 6,655 expressions. JHMDB-Sentences includes 928 videos and each video only has one referring expression.

**Metrics.** We adopt region similarity $\mathcal{J}$, counter accuracy $\mathcal{F}$ and the average value of the two $\mathcal{J}\&\mathcal{F}$ as evaluation metrics for Ref-Youtube-VOS and Ref-DAVIS17. Since the annotations for the validation set of Ref-Youtube-VOS are not publicly accessible, we submit our segmentation results to the official server for evaluation[1]. Our predictions on Ref-DAVIS17 are evaluated using the official evaluation code[2]. We employ mean IoU, overall IoU, Presion@K (K $\in$ [5,6,7,8,9]) and mAP over 0:50:0.05:0.95 as our evaluation metrics on A2D-Sentences and JHMDB-Sentences.

### 4.2 Implementation Details

*4.2.1 Model Settings.* We first introduce the backbones we used. Our training method takes several frames at once and there are on necessary for these frames to have temporal connections. So Video Swin Tranformer [20] with temporal modeling ability is not suitable for our model. We just use ResNet-50 [5] and Swin Transformer [19] as our visual backbones to extract visual features. The output features of the last three layers are used as the visual embedding. Noticed that with these two lighter backbones, we achieve better performance, which is detailed shown in Sec.4.3. As for the text backbone, we choose RoBERTa [18] as our encoder and freeze its parameters in the training stage. We utilize the Transformer with 4 encoder layers and 4 decoder layers and the hidden dimension is 256. The number of position indicators is set as 5.

*4.2.2 Training Details.* We perform downsampling on all frames and their key frames to ensure that the size of the short edge is at least 320 and the size of the long edge is at most 576, fitting GPU memory. During training, we use AdamW as our optimizer. For a fair comparison with previous works, we pre-train our model on Ref-COCO [34] as other works did and then fine-tune it for 6 epochs, the learning rate is initialed as $1e-5$ and decays divided by 10 at the 3rd and 5th epoch. The coefficients for losses are $\lambda_{cls} = 2$, $\lambda_{box} = 5$, $\lambda_{mask} = 2$.

*4.2.3 Inference Details.* During inference, we process each video frame by frame. Every time, our model receives a frame, its tracking information and the referred language expression as input, outputs the predicted binary segmentation mask without post-process.

### 4.3 Main Results

*4.3.1 Ref-Youtube-VOS & Ref-DAVIS17.* We compare our method with state-of-the-art models on the Ref-Youtube-VOS dataset as shown in Tab.1. Among previous methods, PMINet [3] and CITD [15] are the top 2 solutions using ensemble models in the 2021 Ref-Youtube-VOS Challenge. It can be seen that our TF$^2$ with backbone ResNet-50 achieves the overall $\mathcal{J}\&\mathcal{F}$ of 64.6%, which is 5.0% higher than the previous state-of-the-art TCE-RVOS with the same backbone. Previous methods like MTTR [1], ReferFormer [29] and OnlineRefer [28] also use the spatio-temporal backbone Video-Swin-Transformer, which has strong ability to capture both the spatial and temporal clues. Noticed that our TF$^2$ with backbone ResNet-50 even beats the spatio-temporal-based backbone models,

---

[1]https://codalab.lisn.upsaclay.fr/competitions/13520
[2]https://github.com/davisvideochallenge/davis2017-evaluation

**Table 1: Comparison with the state-of-the-art methods on Ref-Youtube-VOS and Ref-DAVIS17.**

| Method | Backbone | Ref-Youtube-VOS | | | Ref-DAVIS17 | | |
| | | $\mathcal{J}\&\mathcal{F}$ | $\mathcal{J}$ | $\mathcal{F}$ | $\mathcal{J}\&\mathcal{F}$ | $\mathcal{J}$ | $\mathcal{F}$ |
|---|---|---|---|---|---|---|---|
| CMSA [32] | ResNet-50 | 34.9 | 33.3 | 36.5 | 34.7 | 32.2 | 37.2 |
| CMSA + RNN [32] | ResNet-50 | 36.4 | 34.8 | 38.1 | 40.2 | 36.9 | 43.5 |
| URVOS [25] | ResNet-50 | 47.2 | 45.3 | 49.2 | 51.5 | 47.3 | 56.0 |
| ReferFormer [29] | ResNet-50 | 55.6 | 54.8 | 56.5 | 58.5 | 55.8 | 61.3 |
| OnlineRefer [28] | ResNet-50 | 57.3 | 55.6 | 58.9 | 59.3 | 55.7 | 62.9 |
| TCE-RVOS [7] | ResNet-50 | 59.6 | 58.3 | 60.8 | 59.4 | 56.5 | 62.4 |
| TF$^2$ (Ours) | ResNet-50 | **64.6** | **62.6** | **66.6** | **65.3** | **62.9** | **67.8** |
| PMINet + CFBI [3] | Ensemble | 54.2 | 53.0 | 55.5 | - | - | - |
| CITD [15] | Ensemble | 61.4 | 60.0 | 62.7 | - | - | - |
| MTTR ($\omega = 12$) [1] | Video-Swin-Tiny | 55.3 | 54.0 | 56.6 | - | - | - |
| ReferFormer ($\omega = 5$) [29] | Video-Swin-Tiny | 59.4 | 58.0 | 60.9 | - | - | - |
| ReferFormer ($\omega = 5$) [29] | Video-Swin-Base | 62.9 | 61.3 | 64.6 | 61.1 | 58.1 | 64.1 |
| OnlineRefer [28] | Swin-Large | 63.5 | 61.6 | 65.5 | 64.8 | 61.6 | 67.7 |
| TF$^2$ (Ours) | Swin-Tiny | 65.7 | 63.6 | 67.8 | 66.3 | 63.4 | 69.2 |
| TF$^2$ (Ours) | Swin-Large | **66.2** | **64.0** | **68.4** | **67.0** | **64.2** | **69.8** |

**Table 2: Comparison with the state-of-the-art methods on A2D-Sentences.**

| Method | Backbone | Precision | | | | | IoU | | mAP |
| | | P@0.5 | P@0.6 | P@0.7 | P@0.8 | P@0.9 | Overall | Mean | |
|---|---|---|---|---|---|---|---|---|---|
| Hu *et al.* [6] | VGG-16 | 63.3 | 35.0 | 8.5 | 0.2 | 0.0 | 54.6 | 52.8 | 17.8 |
| Gavrilyuk [4] | I3D | 69.9 | 46.0 | 17.3 | 1.4 | 0.0 | 54.1 | 54.2 | 23.3 |
| CMSA + CFSA [33] | ResNet-101 | 76.4 | 62.5 | 38.9 | 9.0 | 0.1 | 62.8 | 58.1 | - |
| ACAN [27] | I3D | 75.6 | 56.4 | 28.7 | 3.4 | 0.0 | 57.6 | 58.4 | 28.9 |
| CMPC-V [17] | I3D | 81.3 | 65.7 | 37.1 | 7.0 | 0.0 | 61.6 | 61.7 | 34.2 |
| ClawCraneNet [14] | ResNet-50/101 | 88.0 | 79.6 | 56.6 | 14.7 | 0.2 | 64.4 | 65.6 | - |
| MTTR ($\omega = 10$) [1] | Video-Swin-Tiny | 93.9 | 85.2 | 61.6 | 16.6 | 0.1 | 70.1 | 69.8 | 39.2 |
| Referformer ($\omega = 5$) [29] | Video-Swin-Base | 96.2 | 90.2 | 70.2 | 21.0 | 0.3 | 73.0 | 71.8 | 43.7 |
| OnlineRefer ($\omega = 5$) [28] | Video-Swin-Base | 96.1 | 90.4 | 71.0 | 21.9 | 0.2 | 73.5 | 71.9 | - |
| TF$^2$ (Ours) | Swin-Tiny | **97.6** | 94.0 | 73.6 | 23.2 | **0.3** | 75.8 | 74.0 | 46.2 |
| TF$^2$ (Ours) | Swin-Base | 97.4 | **95.4** | **73.9** | **24.1** | **0.3** | **76.5** | **74.6** | **47.8** |

**Table 3: Comparison with the state-of-the-art methods on JHMDB-Sentences.**

| Method | Backbone | Precision | | | | | IoU | | mAP |
| | | P@0.5 | P@0.6 | P@0.7 | P@0.8 | P@0.9 | Overall | Mean | |
|---|---|---|---|---|---|---|---|---|---|
| Hu *et al.* [6] | VGG-16 | 34.8 | 23.6 | 13.3 | 3.3 | 0.1 | 47.4 | 35.0 | 13.2 |
| Gavrilyuk [4] | I3D | 47.5 | 34.7 | 21.1 | 8.0 | 0.2 | 53.6 | 42.1 | 19.8 |
| CMSA + CFSA [33] | ResNet-101 | 48.7 | 43.1 | 35.8 | 23.1 | 5.2 | 61.8 | 43.2 | - |
| ACAN [27] | I3D | 55.7 | 45.9 | 31.9 | 16.0 | 2.0 | 60.1 | 49.0 | 27.4 |
| CMPC-V [17] | I3D | 65.5 | 59.2 | 50.6 | 34.2 | 9.8 | 65.3 | 57.3 | 40.4 |
| ClawCraneNet [14] | ResNet-50/101 | 70.4 | 67.7 | 61.7 | 48.9 | 17.1 | 63.1 | 59.9 | - |
| MTTR ($\omega = 10$) [1] | Video-Swin-Tiny | 75.4 | 71.2 | 63.8 | 48.5 | 16.9 | 72.0 | 64.0 | 46.1 |
| Referformer ($\omega = 5$) [29] | Video-Swin-Base | 83.1 | 80.4 | 74.1 | 57.9 | 21.2 | 78.6 | 70.3 | 55.0 |
| OnlineRefer ($\omega = 5$) [28] | Video-Swin-Base | 83.1 | 80.2 | 73.4 | 56.8 | 21.7 | 79.6 | 70.5 | - |
| TCE-RVOS [7] | Video-Swin-Base | 83.3 | 80.6 | 74.6 | 58.6 | 22.2 | 78.4 | 70.5 | 56.0 |
| TF$^2$ (Ours) | Swin-Tiny | 87.2 | 82.4 | 78.0 | 60.2 | 25.4 | 85.1 | 76.6 | 60.4 |
| TF$^2$ (Ours) | Swin-Base | **88.3** | **83.0** | **81.2** | **61.2** | **26.9** | **87.6** | **79.7** | **63.0** |

outperforming all the previous works. Additionally, we use the strong Swin-Transformer as the backbone for obtaining more excellent performance. The $\mathcal{J}\&\mathcal{F}$ achieves higher 66.2% with Swin-Base backbone, sufficiently improving the superiority of our method.

Considering that Ref-DAVIS17 only contains 90 videos which is not suitable for training, similar to previous works, we directly use

the model trained on Ref-Youtube-VOS to verify the generality of our TF$^2$. The results are shown in Tab.1. Our model also achieves the state-of-the-art just with backbone ResNet-50. The overall $\mathcal{J}\&\mathcal{F}$ achieves 65.3%, which outperforms previous TCE-RVOS 5.9%. The

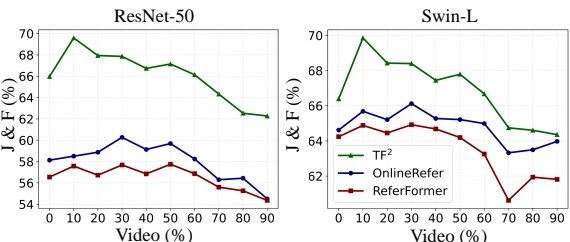

**Figure 5: Frame-wise comparison on the validation set of Red-Youtube-VOS.**

performance has further improvement with the stronger Swin-Transformer with the overall $\mathcal{J}\&\mathcal{F}$ is 67.0%. The results on Ref-DAVIS17 demonstrate the generality of our model.

TF$^2$ concentrates on segmenting the object based on the high-quality tracking information and performs sequential inference starting from key frames. The effectiveness is further proved through the frame-wise comparison with the recent works ReferFormer [29] and OnlineRefer. The results are shown in Fig.5.

*4.3.2 A2D-Sentences & JHMDB-Sentences.* We further compare our method with previous approaches on A2D-Sentences. Previous methods with good performance like ReferFormer [29] and OnlineRefer [28] choose to use the spatio-temporal backbone Video-Swin-Transformer to extract valid visual features. As the characteristic of our training method utilizing ground-truth masks to build tracking information for each frame, there is no temporal information between frames we need to capture. So the Video-Swin-Transformer is not suitable for our model. We just use the more simple spatial Swin-Transformer as our backbone and reach the state-of-the-art. Our method achieves 47.8 mAP which exceeds 43.7% by +1.8% over the previous best result.

The precision results surpass previous methods obviously, which significantly verifies the effectiveness of our model. Among them, the performance with Swin-Tiny backbone exceeds a little with Swin-Base on P@0.5. This is because our method concentrates on spreading valid tracking information rather than modeling the visual information in a more detailed and comprehensive manner, so the performance with Swin-Tiny backbone is close to that with Swin-Base. Besides, noticed that all methods including ours produce low results on P@0.9. Based on previous work, we analyze that this is due to the fact that labels are not accurately labeled by humans, but rather generated by a coarse puppet model.

Following previous works, we use the model trained on A2D-Sentences directly to JHMDB-Sentences without fine-tuning to further demonstrate the generality of our model. The results are shown in Tab.3. Our method also achieves the state-of-the-art on JHMDB-Sentences over each metric.

## 4.4 Discussion
*4.4.1 Components Ablation.* To verify the effectiveness of each component in our model, we conduct ablation studies on Ref-Youtube-VOS using ResNet-50 as the visual backbone. We first remove the selection of key frames in the inference stage and perform simple sequential reasoning from the first frame. As illustrated

**Table 4: Ablation results on Ref-Youtube-VOS with ResNet-50 as the visual backbone, where "KF" denotes the key frame and "TI" denotes tracking information.**

| Components | $\mathcal{J}\&\mathcal{F}$ | $\mathcal{J}$ | $\mathcal{F}$ |
|---|---|---|---|
| Full Model | **64.6** | **62.6** | **66.6** |
| w/o KF | 60.0 ($\downarrow$4.6) | 58.8 ($\downarrow$3.8) | 61.2 ($\downarrow$5.4) |
| w/o KF & TI | 55.5 ($\downarrow$9.1) | 54.4 ($\downarrow$8.2) | 56.7 ($\downarrow$9.9) |

**Table 5: Results of different numbers (*i.e.*, No.) of initialized position indicators (left) and key frames (right) with ResNet-50 as backbone on Ref-Youtube-VOS. $\mathcal{T}$ denotes the time efficiency of the inference process and "R" represents a randomly selected frame as the key frame.**

| No.P | $\mathcal{J}\&\mathcal{F}$ | $\mathcal{J}$ | $\mathcal{F}$ | No.K | $\mathcal{J}\&\mathcal{F}$ | $\mathcal{J}$ | $\mathcal{F}$ | $\mathcal{T}$ |
|---|---|---|---|---|---|---|---|---|
| 1 | 61.5 | 60.2 | 62.8 | R | 56.6 | 55.0 | 58.1 | 1$\times$ |
| 3 | 62.8 | 60.9 | 64.6 | **1** | **64.6** | **62.6** | **66.6** | 1$\times$ |
| **5** | **64.6** | **62.6** | **66.6** | 3 | 58.6 | 57.4 | 59.8 | 2.65$\times$ |
| 8 | 64.0 | 62.1 | 65.9 | 5 | 55.9 | 54.8 | 57.0 | 4.36$\times$ |

in Tab.4, the performance has a significant decrease without the selection of key frames. This reflects the important role of key frames in constructing accurate tracking information. In other words, although the model has obtained a great ability to segment based on accurate key frames, if the key frames constructed during the inference process are not accurate, the performance of the model cannot be fully applied and is still easily prone to error accumulation. We further remove tracking information to offer a deep insight into our model. In these circumstances, position indicators are only guided by language expressions and the inference is also in a parallel way as the same as training. As shown in the last row of Tab.4, there is a further significant decrease in the segmentation effect. This strongly validates the effectiveness of our core idea, *i.e.,* maximizing the accuracy of the inter-frame interaction information.

*4.4.2 Initialized Position Indicators.* The number of initialized position indicators is an adjustable parameter and we explore its value setting. On the one hand, considering that there is only one target object in RVOS, it may seem that setting one position indicator is the best choice. On the other hand, more indicators could provide more instance candidates. We conduct ablation experiments to find the most suitable value for our model. The results are shown in Tab.5. It can be seen that although using only one position indicator can achieve considerable results, the performance further increases with more position indicators for segmenting. The performance of our model saturates when the number of position indicators reaches 5. This is because there is only one positive sample in each frame, more indicators for object detection result in imbalanced label allocation. So we set the number of initialized queries as 5.

*4.4.3 Key Frames Variations.* We conduct a series of variant experiments on the number of key frames to explore its impact on performance. The results are shown in Tab.5. Firstly, we randomly select one frame as the key frame, which plays as a baseline to verify the effectiveness of our selection approach that utilizes the similarity scores between frames and language expressions. We conduct inference based on a random key frame five times and adopt the

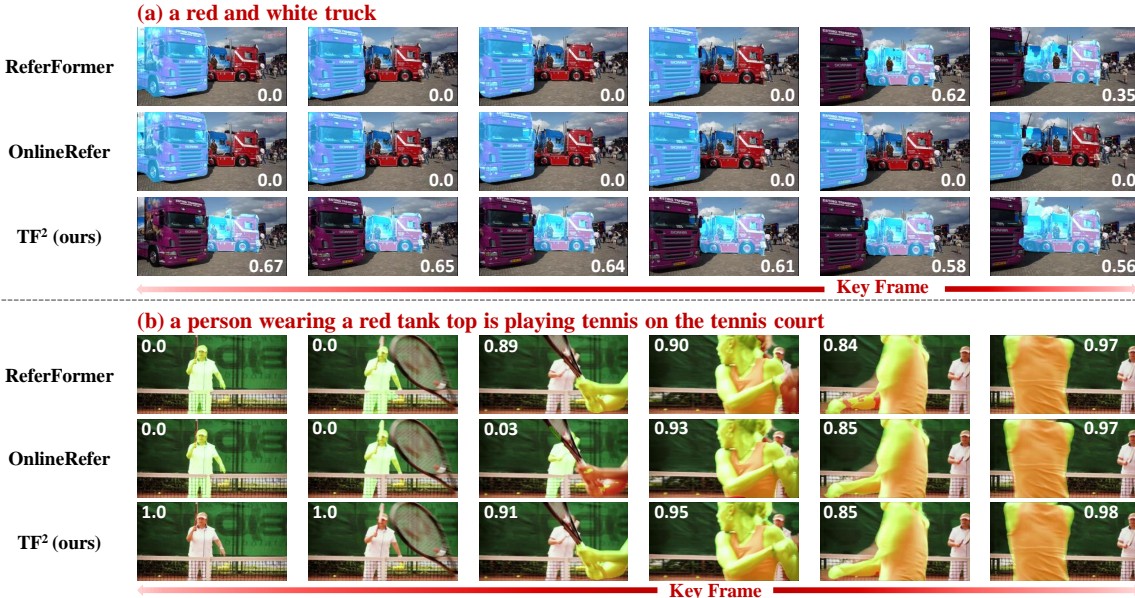

**Figure 6: Visualization of the results from our TF$^2$, ReferFormer [29] and OnlineRefer [28]. Due to the large number of frames in videos, we extract clips to show the segmentation results. We also display the $\mathcal{J}$ score of each frame.**

average value as the final result. Due to the unreliable random key frame, the performance reduces significantly compared to our multimodal similarity-based selection. Considering more key frames as starting points can accelerate the inference process, we further adopt variations with more key frames. However, some selected key frames may not have enough information about the target object referred to by language expressions, resulting in decreases in tracking information quality and segmentation performance. Although the inference could be completed faster, the performance can not be guaranteed. We finally set the number of key frames as 1.

*4.4.4 Training Efficiency.* For a video of $t$ frames, the processing steps required by the clip-level method are about $t/l$, where $l$ is the clip selection length. For the frame-level method, the processing length is $t$. Our method can realize parallel training between frames, with only 1 processing step, thus improving the training efficiency significantly. Specific quantitative results are shown in the Supplementary Material.

## 4.5 Case Study

We analyze cases to show the effectiveness of our method intuitively compared with two recent works, ReferFormer and OnlineRefer, which belong to clip-level methods and frame-level methods respectively. To make a fair comparison, the visual backbone in each model is set as ResNet-50. Segmentation results are shown in Fig.6. The referred object in **case(a)** is the red truck on the right. However, in the first few frames, the purple truck is relatively larger than the red one, and their colors are similar, attracting more attention during segmentation. ReferFormer [29] segments a clip in parallel and lacks direct information interactions between frames. First predicted masks are incorrect under the influence of the purple truck, while the latter masks tend to be accurate. As for OnlineRefer [28] which segments frame by frame, it is negatively affected by the

purple truck. Due to its query propagation mechanism between frames, errors in the first few frames accumulate, leading to the subsequent segmentation errors. Our method chooses the 5th frame as the key frame which is least affected by the left purple truck and performs the segmentation in a spreading way. Benefiting from the accurate tracking information, our TF$^2$ achieves better performance. In **case(b)**, the target person in red moves out of frames sometimes, increasing the segmentation difficulty. For frames where the person appears obviously, ReferFormer [29] segments accurately. While it can not segment the person in other frames correctly. The error accumulation in OnlineRefer [28] is also reflected. In our TF$^2$, we select the 4th frame as the key frame first. With the accurate tracking information sourced form the key frame, the second frame only with a partial target person can also be segmented correctly. It can be seen that the overall segmentation results of all frames achieve better performance than previous works through the bidirectional inference. This intuitively verifies the effectiveness of our utilization of accurate tracking information for promoting segmentation.

## 5 CONCLUSION

In this paper, we propose a tracking-forced framework for referring video object segmentation. It introduces ground-truth masks to construct tracking information, which breaks the dependence of current input on previous output in existing frame-level methods, allows our model to focus on accurately segmenting, and achieves higher training efficiency by parallel frame processing. Besides, to be compatible with the model we trained, we adopt a key-frame-guided sequential inference approach to maximize the accuracy of tracking information to achieve good performance. We conduct extensive experiments on 4 widely used datasets and achieve state-of-the-art performance on these four benchmarks.

## ACKNOWLEDGEMENTS

This research is supported by the National Natural Science Foundation of China (No. 62302243, 62272250), the Natural Science Foundation of Tianjin, China (No. 22JCQNJC01580, 22JCJQJC00150), and the Fundamental Research Funds for the Central Universities, Nankai University (63241442).

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
