# OpenReview forum: "Tracking-forced Referring Video Object Segmentation"
_acmmm.org/ACMMM/2024/Conference — MM2024 Poster_

### Official Review · Reviewer_a3o2 · 2024-05-23

**Rating:** 5
**Confidence:** 4

**Summary:**

This work propose to  a novel  tracking-forced framework to forcing the model to achieve accurate segmentation. The method is novel and performance is competitive.

**Strengths:**

1. The TF2 is novel and performance is promising.
2. Key frame selection is novel.
3. Writing is easy to follow.

**Limitations:**

1. The comparison on A2D-Sentences and JHMDB-Sentences is are not very intuitive. The result with ResNet-50 as backbone is required.
2. Author should provide the inference speed comparison with ReferFormer and OnlineRefer is needed. Besides, the accurate inference time with more key frames is needed instead of a speed up rate.
3. If possible, author should provide the GPU usage during inference and compare it with ReferFormer and OnlineRefer.
3. More visualization results of the attention map in decoder should be provided to show that the pervious frame tracking can result in more accurate attention map.

**Suitability:**

3

---

### Official Review · Reviewer_3cmL · 2024-05-23

**Rating:** 4
**Confidence:** 3

**Summary:**

This paper proposes a tracking-forced framework for the task of RVOS. The framework leverages ground-truth masks to construct tracking information, mitigating the dependence of current input on previous output prevalent in existing frame-level methods. This design enables the model to focus solely on accurate segmentation, while enhancing training efficiency through parallel frame processing. Furthermore, to ensure compatibility with the model, a key-frame-guided sequential inference approach is adopted, maximizing the accuracy of tracking information to achieve optimal performance. Extensive experiments on four widely-used datasets demonstrate that the proposed method achieves state-of-the-art results.

**Strengths:**

The paper provides a clear definition of RVOS and presents a well-structured and moderately readable narrative, complemented by high-quality figures and charts.

The authors conducted comprehensive comparisons across four datasets to validate the performance advantages of their proposed method.

**Limitations:**

Certain aspects of the paper require further clarification and refinement:
1. Fig. 1c seems inconsistent with the statement in line 219, where the authors mention "We first train our TF2 in a parallel way and then conduct sequential inference." The authors should clarify this discrepancy rigorously.

2. The term "tracking information" may be misleading, as TF2 solely utilizes the mask from the previous frame, but "tracking information" intuitively gives the impression that you are using some motion information (e.g. Kalman filtering, which is commonly used in multi-object tracking) in the model. A more descriptive terminology would be beneficial.

3. The authors should clarify whether the use of CLIP for key frame determination causes any potential mismatch or incompatibility with TF2.

4. Section 3.2.1, particularly the part following Equation 4, seems confusing. While Equation 4 suggests a single key frame per video, line 450 implies multiple key frames. The authors should refine the description to enhance reader comprehension.

5. Equation 5 requires further explanation, such as the details of TF2(,,).

6. While TF2 achieves state-of-the-art performance on the four datasets, a significant portion of this improvement stems from the key frame selection trick, which could potentially be applied to other methods as well. Moreover, this key frame selection effectively renders TF2 an offline method. The authors should specify the mode (online, semi-online, or offline) of other compared methods for a fair comparison between online and offline approaches.
7. The authors should explicitly state that the proposed method is offline and carefully analyze the potential impact of the key frame trick on the community.

8. A final careful check is recommended before submission, as line 133 contains a repeated word ("decoupling").


If I were to design TF2, instead of using CLIP for key frame selection, an alternative approach could involve integrating it directly into TF2, i.e., first using TF2 to get the masks of all the frames and the matching degree of descriptive text with each frame, then selecting the key frame based on the highest matching degree, and finally propagating the video in both directions from the key frame to refine those previous masks. This would eliminate the perceived "trick" aspect and allow the authors to emphasize the offline nature of their method, clearly differentiating it from previous online approaches.

While the paper presents an innovative approach and achieves state-of-the-art results, the authors should address the limitations outlined above to enhance the clarity, rigor, and fairness of their work. In particular, explicit acknowledgment of the offline nature of their method and a candid analysis of the potential impact of the key frame trick are crucial. Depending on the authors' response to these concerns, further adjustments to the review decision may be warranted.

**Suitability:**

2

---

### Official Review · Reviewer_BokA · 2024-05-24

**Rating:** 4
**Confidence:** 3

**Summary:**

The paper addresses the problem of Referring Video Object Segmentation (RVOS), a task that segments objects in videos based on natural language descriptions. Traditional methods either process multiple frames in parallel, lacking inter-frame interactions, or process frames sequentially, leading to error accumulation. The authors propose a novel tracking-forced framework (TF2) that introduces high-quality tracking information from ground-truth masks of previous frames to guide the segmentation process, improving segmentation accuracy and training efficiency. The proposed method outperforms existing methods on widely used benchmarks.

**Strengths:**

The paper is well written.
The introduction of the tracking-forced framework is a significant contribution, addressing the limitations of both clip-level and frame-level methods.
By decoupling the previous output from the current input, the proposed method allows for parallel training, which enhances training efficiency.
The method is thoroughly evaluated on several datasets, and achieves state-of-the-art performance on multiple benchmarks, including Ref-Youtube-VOS, Ref-DAVIS17, A2D-Sentences, and JHMDB-Sentences.

**Limitations:**

The reliance on ground-truth masks for constructing tracking information may limit the method's applicability. The process of selecting key frame based on semantic similarity add complexity and computational overhead during the inference stage, which seems to be irremovable. During inference, the segmentation correctness of the key frame is not guaranteed, which may result in error segmentation of the whole video, since the model is trained with the assumption that the guidance of previous frames is the ground truth.
While the method benefits from the above assumption, it may not fully exploit temporal information compared to methods designed with strong temporal modeling capabilities.

**Suitability:**

3

---

### Meta-Review · Area_Chair_yBcp · 2024-07-01

**Recommendation:** Accept (Poster)
**Confidence:** 5

**Metareview:**

The reviewers unanimously recommend acceptance. After checking the rebuttal and the paper, the AC agrees with this assessment.